# Advancements in TGF-β Targeting Therapies for Head and Neck Squamous Cell Carcinoma

**DOI:** 10.3390/cancers16173047

**Published:** 2024-08-31

**Authors:** William R. Britton, Isabel Cioffi, Corinne Stonebraker, Matthew Spence, Ogoegbunam Okolo, Cecilia Martin, Brian Henick, Hiroshi Nakagawa, Anuraag S. Parikh

**Affiliations:** 1Herbert Irving Comprehensive Cancer Center, Columbia University, New York, NY 10032, USA; wrb2120@cumc.columbia.edu (W.R.B.);; 2Columbia Vagelos College of Physicians and Surgeons, Columbia University, New York, NY 10032, USA; 3Organoid and Cell Culture Core, Columbia University Digestive and Liver Diseases Research Center, Columbia University, New York, NY 10032, USA; 4Division of Digestive and Liver Diseases, Department of Medicine, Columbia University, New York, NY 10032, USA; 5Department of Otolaryngology-Head and Neck Surgery, Columbia University, New York, NY 10032, USA

**Keywords:** head and neck squamous cell carcinoma, TGF-β, epithelial-to-mesenchymal transition, bintrafusp alfa, BCA-101, dalantercept, LY3200882, SHR-1701

## Abstract

**Simple Summary:**

TGF-β is an important cytokine shown to drive oncogenesis in head and neck squamous cell carcinoma (HNSCC) through its diverse influences on the tumor microenvironment. While this cytokine is vital in maintaining tissue homeostasis in normal head and neck epithelia, in cancer, it paradoxically drives metastasis, angiogenesis, immune evasion, and therapy resistance. Despite promising preclinical data, the outcome of clinical trials of TGF-β inhibitors for HNSCC has been suboptimal. Patient stratification is warranted to improve this targeted therapy.

**Abstract:**

Head and neck squamous cell carcinoma (HNSCC) is the sixth leading cause of cancer worldwide according to GLOBOCAN estimates from 2022. Current therapy options for recurrent or metastatic disease are limited to conventional cytotoxic chemotherapy and immunotherapy, with few targeted therapy options readily available. Recent single-cell transcriptomic analyses identified TGF-β signaling as an important mediator of functional interplays between cancer-associated fibroblasts and a subset of mesenchymal cancer cells. This signaling was shown to drive invasiveness, treatment resistance, and immune evasion. These data provide renewed interest in the TGF-β pathway as an alternative therapeutic target, prompting a critical review of previous clinical data which suggest a lack of benefit from TGF-β inhibitors. While preclinical data have demonstrated the great anti-tumorigenic potential of TGF-β inhibitors, the underwhelming results of ongoing and completed clinical trials highlight the difficulty actualizing these benefits into clinical practice. This topical review will discuss the relevant preclinical and clinical findings for TGF-β inhibitors in HNSCC and will explore the potential role of patient stratification in the development of this therapeutic strategy.

## 1. Introduction

Head and neck squamous cell carcinoma (HNSCC) is the sixth most common cancer diagnosis worldwide, with an estimated 890,000 new cases and over 450,000 deaths each year per the latest GLOBOCAN estimates from 2022. This cancer includes malignant lesions which arise within the oral cavity, nasal cavity, and salivary glands, as well as the pharynx and larynx. Common risk factors for this condition include smoke inhalation, alcohol and areca nut consumption, and human papilloma virus (HPV) infection [1]. In addition, HNSCC is linked to genetic polymorphisms associated with impaired DNA repair, such the FANC genes in Fanconi anemia, as well as with impaired immunity, with variants affecting genes like CTLA4 and IL10 [2,3,4]. Clinical decisions on management are often based on criteria of controversial utility, such as depth of invasion, lymphovascular invasion, and perineural invasion [5,6]. According to the Surveillance, Epidemiology, and End Results (SEER) Program of the U.S. National Cancer Institute released in 2009, the overall HNSCC five-year survival rate in patients from 2002 to 2006 was 65.9%, an improvement from 52.7% between 1992 and 1996 [7]. It is thought that these improvements in survival are likely secondary to the increased prevalence of HPV-positive cancers, rather than an improvement in therapeutic options [7,8]. However, five-year survival within metastatic disease between 2002 and 2006 for oral cavity, tongue, and tonsillar cancers was 29.5%, 35.2%, and 41.5% respectively [7]. This suggests ongoing difficulty in managing advanced disease. 

The mainstay of therapy for HNSCC involves a combination of surgical resection, radiation, and cytotoxic chemotherapy. Chemotherapy typically consists of a platinum-based regimen, which is often accompanied by significant toxicity including, but not limited to, dysphagia, severe leukopenia, granulocytopenia, and muscular fibrosis [9,10]. In early-stage disease with limited nodal involvement, treatment consists of surgery or radiation alone and can have a three-year survival rate of >80% [11]. However, survival quickly drops off in more advanced tumors. In nonresectable stage III or stage IV tumors with no distant hematogenous metastasis, treatment with radiation and bolus cisplatin is associated with a three-year survival rate of around 37% [12]. In the metastatic or recurrent setting, management options may also include immune checkpoint inhibitors such as pembrolizumab or the EGFR antagonist cetuximab, given the upregulation of the EGFR pathway in 80–90% of HNSCC cases [13,14,15]. While clinical trials in metastatic and recurrent HNSCC have demonstrated the superiority of pembrolizumab or nivolumab to standard single agents such as methotrexate, docetaxel, or cetuximab, the objective response is in the range of 13–16% [16,17]. Further, those ineligible for pembrolizumab or other checkpoint inhibitors can be administered cetuximab monotherapy, which has an objective response rate of 11% [18]. As such, the lack of alternative targeted therapies for patients with metastatic or recurrent HNSCC represents an unmet clinical need. 

The current limitations in therapeutic strategies for the treatment of metastatic and recurrent HNSCC are multifactorial and are greatly impacted by the heterogeneous nature of both HPV-positive and HPV-negative HNSCC. Factors implicated in carcinogenesis include common chromosomal alterations in HPV-negative HNSCC such as the loss of 9p21 and 3p21, which contain important tumor suppressor genes CDKN2A and TP53. Furthermore, HPV-positive HNSCC is associated with the disruption of other tumor-suppressing genes, notably p53 and RB1 [19]. Importantly, the heterogeneity of HNSCC is further influenced by a dynamic tumor microenvironment (TME), in which stromal cells hold great importance in pathogenesis. Specifically, cancer-associated fibroblasts (CAFs) are known to secrete important growth factors, immunosuppressive cytokines, and matrix metalloproteinases (MMPs), contributing to processes such as growth, angiogenesis, immune evasion, epithelial-to-mesenchymal transition (EMT), and metastasis [20]. It is the combination of heterogeneous genetic abnormalities and a unique TME that make this disease so difficult to treat. 

To further characterize HNSCC and identify alternative therapeutic targets, our group performed a single-cell transcriptomic analysis of HNSCC [21]. This analysis identified the transforming growth factor-beta (TGF-β) pathway as an important mediator of crosstalk between CAFs and cancer cells. Our data showed that TGF-β may contribute to a signaling program which drives invasiveness, treatment response, and immune evasion [21,22]. This program, termed partial epithelial-to-mesenchymal transition (p-EMT), confers mesenchymal characteristics onto a subset of epithelial cells, localized to the leading edge of a tumor, and is thought to be induced by adjacent CAFs [21,22]. p-EMT represents an intermediary state along a spectrum, in which many intercellular connections are maintained while the ECM is restructured, allowing cancer cells to concurrently escape from the stromal scaffold. This process is vital in promoting local tissue invasion and metastasis [21,23]. This program was found to be a key component of intratumoral heterogeneity (ITH), which encompasses the many different subpopulations of malignant and stromal cells thought to drive metastasis and immune evasion. ITH and p-EMT have been linked to poor clinical outcomes in HNSCC [23,24]. These results generate renewed interest in the TGF-β pathway as an alternative therapeutic target for recurrent and metastatic disease and prompt a critical review of previous data underlying the use of TGF-β inhibitors in HNSCC. This review will provide an overview of the role of TGF-β in HNSCC, describe the relevant preclinical and clinical findings of TGF-β inhibitors in HNSCC, and explore the potential role of patient stratification in the development of future therapeutic strategies. 

## 2. TGF-β Signaling

It is first important to understand the unique signaling pathways underlying the impact of TGF-β on cancer progression. TGF-β is a pleiotropic cytokine involved in a range of normal tissue processes such as proliferation, apoptosis, differentiation, and cell migration. This cytokine belongs to the TGF-β superfamily, which comprises other growth factors such as activins, bone morphogenic protein (BMP), anti-Mullerian hormone (AMH), and growth and differentiation factors (GHF) [25]. While this review will focus primarily on the action of TGF-β, recent studies have suggested that many of the ligands within this superfamily additionally impact tumorigenesis, demanding important consideration of methods of therapy resistance to direct TGF-β inhibitors [26]. While there are three ligands within the TGF-β family (TGF-β1, TGF-β2, and TGF-β3) which are highly homologous and perform similar biologic functions, previous studies have shown the upregulation of TGF-β1 in HNSCC epithelia relative to normal tissue, suggesting the heightened importance of this ligand in tumor pathogenesis [27]. 

Regarding the role of this ligand in cancer, it is released by the tumor or by other stromal components of the tumor microenvironment [28]. Inactive (latent) TGF-β cytokines exist in the extracellular matrix, which, upon activation, bind to a constitutively active receptor, TGF-β Receptor Type II (TBRII). Together, this ligand–receptor pair forms a heterotetrameric complex with TGF-β Receptor Type I (TBRI), also known as activin receptor-like kinase, which results in the phosphorylation of TBRI (Figure 1) [28]. Throughout this process, the co-receptor, TGF-β Receptor III, serves to modulate access to ligands of TBRI and TBRII [29]. After phosphorylation, the complex of TBRI and TBRII goes on to induce either canonical or non-canonical signaling.

Through the canonical pathway, TBRI phosphorylates receptor-regulated SMAD proteins, most commonly SMAD2 and SMAD3. These proteins then form heterotrimers with common-mediator SMADs such as SMAD4, which then translocate to the nucleus and influence the transcription of target genes [30,31,32]. At the same time, inhibitory SMADs such as SMAD6 and SMAD7 compete for binding with TGF-β receptors and recruit ubiquitin ligases to degrade receptor-regulated SMADs and TBRI, thereby modulating the signaling intensity of these SMADs [33,34]. TGF-β can also exert its effects through alternative non-canonical signaling pathways independently of SMAD proteins. These pathways include the activation of RhoA GTPase and signal induction through NFKappaB/IL1B, p38 MAPK, and many others [35,36,37,38]. 

## 3. TGF-β in HNSCC

TGF-β acts as either a tumor suppressor or tumor promoter depending on the tissue context. In epithelial tissue and the early stages of oncogenesis, TGF-β acts as a tumor suppressor. This effect is thought to arise from its influence on cell growth and apoptosis [39]. Specifically, in settings of high proliferation, TGF-β inhibits growth through the upregulation of cyclin-dependent kinase (CDK) inhibitors, such as p15 or p21, which inhibit cell cycle progression [40]. TGF-β also leads to the downregulation of c-MYC, a key activator of cell growth and proliferation. [41,42]. TGF-β has similarly been hypothesized to maintain homeostasis by inducing apoptosis. This process is carried out through the SMAD-7-dependent activation of ataxia telangiectasia mutated (ATM), a kinase sensitive to DNA damage, and p53, a key tumor suppressor driving cell apoptosis [43]. Interestingly, mice with disruption in TGF-β signaling, through the conditional KO of TBRI, develop HNSCC, thereby supporting a protective role of TGF-β signaling in normal head and neck epithelial tissue [44]. 

The upregulation of TGF-β in HNSCC may paradoxically promote tumor progression. In fact, using a previously described transgenic mouse model [45] in which an inducer, the glucocorticoid antagonist Ru486, was applied topically to the oral cavity of mice to induce its linked TGF-β1 transactivator, researchers found that increased TGF-β1 signaling in the head and neck epithelium led to inflammation, angiogenesis, and epithelial hyperproliferation [27]. It is important to note that these epithelial changes are likely secondary to the influence of infiltrating inflammatory cells, leading to reactive epithelial changes, thereby counteracting the cell-cycle-suppressive effects of TGF-β. As such, it is possible that TGF-β signaling may promote a pro-oncogenic chronic inflammatory environment early in carcinogenesis through its impact on other cells in the tumor microenvironment. 

The switch from tumor suppression to tumor promotion is thought to be secondary to disruptions in cytostatic pathways, thereby eliminating the tumor-suppressive arm of the TGF-β pathway. Specifically, one study identified a reduction in SMAD-4 expression within metastatic HNSCC, and similarly witnessed the development of HNSCC within mice with conditional SMAD-4 knockout [46]. These results thereby propose a guardian role of SMAD-4 in maintaining genomic stability. Using mRNA transcriptional analysis, studies found a reduction in TGFBRII in HNSCC cancer cells and the concurrent accumulation of extracellular TGF-β1. This increase in TGF-β1 was hypothesized to increase inflammation and angiogenesis through its interaction with the tumor stroma [47]. As such, imbalanced TGF-β signaling may shift the tumor microenvironment to favor growth and metastasis through a unique tumor–stromal interface. 

## 4. TGF-β and the Tumor Microenvironment

Downstream of its canonical and non-canonical pathways, TGF-β interacts with many components of the tumor microenvironment (TME). Much of this signaling takes place across a tumor–stromal interface, in which CAF-derived TGF-β influences EMT, immune evasion, and angiogenesis [25]. It is through these processes that TGF-β plays a vital role in generating a pro-oncogenic TME. 

### 4.1. TGF-β and EMT

Previous studies have shown that TGF-β is a significant contributor to EMT, as summarized in Figure 1 [48,49]. This process is hypothesized to be secondary to the influence of CAF-derived TGF-β on cancer TBRII receptors. This interaction leads to vital modifications in cancer basement membrane-associated proteins. This includes a SNAIL1-dependent reduction in E-cadherin expression or a SNAIL1/SNAIL2-dependent increase in matrix metalloprotein-9 (MMP-9) expression [50,51,52]. TGF-β has also been shown to act synergistically with epidermal growth factor (EGF) to induce EMT. Previous in vitro HNSCC research has shown that co-stimulation with TGF-β and EGF leads to an increase in vimentin, decrease in E-cadherin, and increase in invasiveness, greater than the effect of stimulation with either of these agents alone [49]. This synergistic promotion of EMT may likely be secondary to the impact of EGF and TGF-β on notch-mediated EMT. TGF-β has previously been shown to inhibit Notch-3-mediated cell differentiation and promote a Notch-1-mediated increase in EMT to create cancer-initiating cells [53]. Interestingly, Notch-1 can also induce Notch-3, reflecting its context-dependent role as both a tumor suppressor and tumor promoter [54]. However, the EGF pathway has also been previously implicated as a negative regulator of Notch-1, shown to suppress the cell-differentiation capacity of this pathway [55]. As such, it is likely that TGF-β and EGF work in concert to push notch signaling processes to favor EMT over cell differentiation. 

Importantly, EMT exists along a spectrum, in which a subset of cells retain some epithelial characteristics [56]. TGF-β has been shown to drive this partial phenotype, conferring a more aggressive and locally invasive p-EMT phenotype on to the leading edge of a growing tumor [21]. Specifically, studies have shown that the administration of TGF-β to an in vitro HNSCC cell line (SCC9) induces the expression of a p-EMT program, while the inhibition of TGF-β reduces the expression of this program [21]. Similar effects were observed when manipulating the expression of TGF-β-induced (TGFBI), a key downstream target of TGF-β, suggesting the importance of this target in mediating p-EMT [21]. As such, TGF-β, and its associated downstream targets, play a vital role in inducing this partial phenotype and driving invasiveness and therapy resistance [21,22,24]. 

### 4.2. TGF-β in the Immune Microenvironment

TGF-β has also been shown to play an important immunosuppressive role in the immune–tumor interface, as summarized previously and depicted in Figure 2. Briefly, exposure to TGF-β has been shown to favor the differentiation of CD4+ T cells into suppressive regulatory T cells [57]. This finding is supported by clinical data showing the enrichment of regulatory T cells and the depletion of cytotoxic T cells in peripheral blood samples of HNSCC patients [58,59]. Within the innate immune system, TGF-β impairs dendritic cell (DC) maturation and mobility [60]. This is also reflected within clinical data, as patients with HNSCC are shown to have increased DC inactivation [61]. Additionally, TGF-β influences the differentiation of macrophages, favoring an anti-inflammatory M2 phenotype [62], known to be associated with an increased pathologic grade in HNSCC [63]. The influence of TGF-β on neutrophil infiltration is context dependent. Previous studies have suggested that tumor-associated neutrophils (TANs) have differential states of activation, either inhibiting tumor formation (N1 phenotype) or promoting it (N2 phenotype) [64]. TGF-β has been shown to recruit neutrophils that favor a pro-oncogenic environment (N2), which secrete angiogenic factors and matrix-degrading enzymes [64]. 

It is additionally important to examine the role of TGF-β in the unique immune profile of patients with HPV+ HNSCC. Despite the favorable prognosis of HPV-positive disease over HPV-negative disease, studies found increased expression of TGF-βR1 and increased related downstream noncanonical P38-MAPK signaling within HPV-positive HNSCC and cervical cancer [65]. Further, in stratifying HPV-positive HNSCC, studies have shown that cancers with high expression of E6, an important HPV-related viral oncogene, are linked to a poorer prognosis and increased risk of distant metastasis. Gene enrichment analyses identified TGF-β as one of the most highly upregulated transcripts in tumors with elevated expression of E6, suggesting a unique connection between this cytokine and viral pathogenesis [66]. However, the underlying molecular mechanistic relationship between TGF-β and E6 seropositivity has yet to be fully elucidated. It is possible that an increase in TGF-β may alter the TME to favor metastasis and immune evasion, leading to persistent viral disease and poor tumor prognosis.

### 4.3. TGF-β and Angiogenesis

Finally, TGF-β signaling is significantly implicated in tumor angiogenesis; however, its role in this process is highly variable. The angiogenic effects of TGF-β are heavily dependent on the tissue context and are mediated through ALK-1- or ALK-5-dependent processes (Figure 3) [67]. Within endothelial cells, the binding of ALK-1 by TGF-β or BMP-9/-10 and the resulting activation of SMAD -1, -5, or -8 leads to the induction of angiogenesis [68,69]. On the other hand, the binding of TGF-β to the ALK-5 receptor leads to the activation of SMAD-2 or -3, resulting in the inhibition of angiogenesis and the inactivation of ALK-1 [68]. It is important to note that some studies have shown paradoxically anti-angiogenic effects from BMP-9 through its action on ALK-1 [70,71]. Specifically, BMP-9 administration to in vitro endothelial cells inhibited basic fibroblast growth factor (bFGF)-induced migration and VEGF-induced angiogenesis through the ALK1 receptor, suggesting an antiangiogenic role of this ligand [71]. However, other studies using a soluble chimeric protein for ALK-1, with high affinity to BMP-9 and BMP-10, blocking this signaling pathway, showed greatly reduced VEGF- and FGF-induced vessel formation in a chorioallantoic membrane assay [72]. This same ligand trap was shown to reduce tumor microvascular density in a mouse xenograft model of HNSCC [73]. As such, the impact of ALK-1 on angiogenesis appears to be context dependent, and further studies are needed to clarify the tissue-specific role of TGF-β and ALK-1/5 signaling within HNSCC.

## 5. TGF-β Inhibitors in HNSCC

Given the important role of TGF-β in tumorigenesis, many attempts have been made to develop TGF-β targeting therapies in HNSCC. To date, there have been several clinical trials in HNSCC utilizing TGF-β targeting agents either in monotherapy or in combination with other cytotoxic regimens or immunotherapies (Table 1). The remainder of this review will focus on these trials, which highlight the use of small-molecule inhibitors, bifunctional agents, and activin-like receptor antagonists to target various components of the TGF-β pathway in cancer.

### 5.1. Small-Molecule Kinase Inhibition

Some early studies used small-molecule inhibitors (SMIs) to target TGF-β in HNSCC. These inhibitors have a molecular weight of less than 900 Da and are used to target specific proteins within cancer cells or the tumor microenvironment. Advantages of these agents include higher selectivity, convenient drug intake, and favorable tolerability [74]. While substantial progress has been made in the use of SMIs in the treatment of cancer, the development of these agents for HNSCC is relatively novel and ongoing. Three TGF-β targeting drug candidates have been investigated for their utility in the treatment of HNSCC: galunisertib, vactoserib, and LY3200882. All three drug candidates serve as ATP-competitive inhibitors of the intracellular serine–threonine kinase domain of TGFBR1. The binding of these SMIs prohibits the phosphorylation of TGFBR1 and thus inhibits the initiation of downstream canonical and non-canonical TGF-β pathways. However, among these drugs, only LY3200882 proceeded to clinical trials for HNSCC [75]. A closer look at pre-clinical outcomes sheds light on the efficacy and clinical trajectory of these targeted therapies. 

**Table 1 cancers-16-03047-t001:** Results from completed and ongoing clinical trials of TGF-β signaling inhibitors.

Agent	Study NCT Number and [Article Citation]	Phase	Status	Study Population	Methods	Number of Participants	Number of Participants with HNSCC	Clinical Efficacy HNSCC	Most Frequent Grade 3 or above TRAES
Bintrafusp alfa	NCT02517398 [76]	I	Completed	Refractory, recurrent, or metastatic HNSCC	Monotherapy	32	32	PR: 4/32 (13%) SD 4/32 (13%)	Increased liver enzymes Anemia
	NCT03427411 [77]	II	Completed	Refractory, recurrent, or metastatic HPV-associated malignancies	Monotherapy	59	15	PR: 4/15 (27%) DPR: 2/15 (13%)	Anemia Keratoacanthoma
	NCT04247282 [78]	I/II	Completed	HPV-unrelated resectable HNSCC	Neoadjuvant therapy with resection	14	14	PR: 5/14 (35.7%)	Vasculitis
SHR-1701	NCT03710265 [79]	I	Unknown status	Metastatic or locally advanced solid tumors	Monotherapy	171	10	PR: 2/10 (20%) SD: 4/10 (40%)	Increased liver enzymes Anemia
BCA-101	NCT04429542 [80]	I	Recruiting	Refractory advanced solid tumors	Monotherapy/combination therapy with pembrolizumab	BCA-101: 40 BCA101 + pembrolizumab: 13	BCA-101: 6 BCA-101 + pembrolizumab: 7	BCA-101 + pembrolizumab: PR: 2/7 (28%)	Anemia Hematuria
LY3200882	NCT02937272 [75]	I	Active	Advanced Solid tumors (treatment naïve)	Combination therapy with CT/RT	139	3	CR: 1/3 (33%) SD: 2/3 (66%)	Myelosuppression
Dalantercept	NCT00996957 [81]	I	Completed	Advanced solid tumors	Monotherapy	29	3	PR: 1/3 (33%) PSD: 1/3 (33%)	Congestive heart failure Anemia
	NCT01458392 [82]	II	Completed	Refractory, recurrent, or metastatic HNSCC	Monotherapy	46	46	PR: 2/40 (5%) SD: 14/40 (35%)	Hyponatremia Amylase/lipase elevation

PR, partial response; SD, stable disease; CR, complete response; DPR, delayed partial response; CT, chemotherapy; RT, radiation therapy, PSD, prolonged stable disease.

Two of these small-molecule inhibitors, galunisertib and vactoserib, were investigated as radiosensitizing agents in HNSCC. Previous results suggest a complex role of TGF-β in radioresistance. While some early studies argue that TGF-β plays a tumor-suppressive role in the post-radiation tumor microenvironment [83], other studies show contrasting results, with TGF-β inhibitors in combination with radiation reducing tumor growth in an in vivo model of breast cancer [84]. Initial studies found that galunisertib and vactoserib significantly reduced cell survival and migration across three different HNSCC lines and that these effects were amplified in combination with radiotherapy [85,86]. While both therapies are currently being investigated in clinical trials for other cancer subtypes, clinical investigation has yet to be started in HNSCC. 

Building on early preclinical data of TGF-β SMIs, a separate group developed LY3200882, which is more potent and selective than galunisertib [87]. This therapy was shown to significantly decrease TGF-β associated SMAD signaling within in vitro tumor cells and displayed promising anti-tumor potential in a mouse model of breast cancer [87]. Given the preclinical success of LY3200882, it was brought to clinical trials within individuals with advanced solid tumors (NCT02937272). This clinical study consisted of three arms (A-C), with A. representing a dose-escalation arm of LY3200882 monotherapy in advanced solid tumors, B. a dose-expansion arm within grade 4 gliomas, and C. a combination treatment arm with LY3200882, a PD-L1 inhibitor LY3300054, cisplatin, and radiotherapy for a cohort of three patients with locoregionally advanced HNSCC. Patients within cohort C. were all male and had a mean age of 52. One patient had previous surgery for his disease, but the others were treatment naive. Within the HNSCC cohort, 1/3 (33%) achieved a complete response, while 2/3 (66%) achieved stable disease, resulting in an objective response rate (ORR) of 33% and disease control rate (DCR) of 100% [75]. While these results seem promising, this study had several important limitations. Most notably, only three participants with HNSCC were included, thereby significantly limiting the power of their findings. Additionally, they also included other first-line therapies for HNSCC in their treatment regimen, thereby limiting their capacity to draw conclusions about the efficacy of the TGF-β-inhibiting component of therapy. 

### 5.2. Joint TGF-β and PDL1 Inhibition

Immune checkpoint inhibitors targeting PD-L1, such as pembrolizumab, have been useful in the treatment of metastatic or recurrent HNSCC. Patient outcomes with these agents alone or in combination with cytotoxic therapies are still underwhelming, with an ORR of less than 20% [88]. However, it is important to note that upon stratifying patients with recurrent or metastatic HNSCC, those with HPV-positive disease often have a much more favorable response, with some studies showing an ORR of >30% for HPV-positive HNSCC treated with pembrolizumab [89]. Resistance to immunotherapy is thought to be secondary to many factors which influence the tumor microenvironment and the capacity to evade the immune system [90]. Specifically, preclinical studies demonstrated that the treatment of in vitro lung tumor cells with exogenous TGF-β led to an increase in the surface expression of PD-L1, thereby providing a means to avoid immune detection [91]. As such, bifunctional agents integrating both PD-L1 blockade with TGF-β antagonism have been developed to overcome therapy resistance. Perhaps the most heavily studied in HNSCC is bintrafusp alfa (BA), which consists of an IgG1 anti-PD-L1 component fused to the extracellular domain of TGF-BRII, shown to effectively bind all three isoforms of TGF-β (Figure 4) [92].

Preclinical studies of BA in cancer showed promising results. Joint PD-L1 and TGF-β inhibition was able to produce robust responses in multiple squamous cell carcinoma and non-small cell lung cancer cells [91,93]. Similarly, this treatment strategy significantly reduced the tumor burden in murine models of breast and colorectal cancer [94]. Another preclinical study used murine models of HNSCC to investigate the stratification of responders vs. non-responders to BA [95]. Using injected LY2 cells [96], cells derived from 4NQO treatment [97], or cells with a SMAD-4 mutation to overexpress TGF-β, this study found a durable reduction in mouse tumor size and identified immune permissive tumor niches that promoted therapy response [95]. Specifically, responders to therapy showed higher tumor infiltration of CD4+, CD8+, and natural killer cells as well as increased markers of antigen presentation and macrophage activation [95]. Taken together, these studies suggest the high potential of BA to produce a sustained therapeutic response within multiple models of HNSCC. 

There have been several major clinical trials examining the therapeutic potential of BA, or similar bifunctional agents, in metastatic, recurrent, or locally advanced/surgically resectable HNSCC. Notably, there was a phase I open-label dose-escalation trial in 32 patients with advanced HNSCC (NCT02517398), which included participants who received prior platinum-based therapy with or without additional cetuximab therapy [76]. No participants had prior T-cell targeting therapy. There were 32 participants in total, with 27 of those being male, with an average age of 60. Participants received treatment every 2 weeks, with a median duration of treatment of 12 weeks, but a range from 2 to 96 weeks in total. There was an objective response rate (ORR) of 13%, with four patients achieving a partial response (PR), and two of these patients had an ongoing response at 16.5+ and 19.6+ months at cutoff. Four additional patients achieved stable disease resulting in a DCR of 34%. Interestingly, among these patients with an objective response, three were HPV+, with a cohort ORR of 3/9 (33%). Due to favorable activity within HPV+ tumors, a phase II clinical expansion trial was conducted in HPV+ HNSCC and cervical SCC, including the prospectively defined HPV+ cohort from NCT02517398 [77]. In the pooled data, 15 HPV+ HNSCC participants were included in the final analysis. Of this cohort, four attained a PR, and two attained a PR after a period of initial progression, raising the ORR to 6/15 (40%). This ORR is slightly higher than what is seen for pembrolizumab monotherapy in HPV-positive metastatic or recurrent HNSCC [89]. While these results suggest that there is an improvement in treatment outcomes for patients treated with BA who have HPV-positive disease, it is also important to interpret these results in the context of the limited study size. 

An alternative agent, containing similar functional components to BA, known as SHR-1701, was shown in preclinical studies to efficiently target PD-L1, TGF-β-1, and TGF-β-3 [79,98]. This agent was similarly studied in patients with advanced metastatic cancer naïve to checkpoint inhibitor treatment, in a multicenter dose-escalation, dose-expansion and clinical expansion trial across 19 centers in China (NCT03710265) [79]. Ten patients within the clinical expansion had HNSCC. Nine of these patients were male, with an average age of 57. Nine of these patients had also undergone prior chemotherapy treatment. These patients were treated with an infusion of SHR-1701 until disease progression, withdrawal, intolerable toxicity, or study completion. Two of these patients achieved a PR (20%) and four achieved stable disease (40%) with an ORR of 20%. Importantly, these patients were not assessed for HPV positivity as 7/10 patients had nasopharyngeal HNSCC, which is unlikely to be caused by HPV. As such, differential response between the HPV-positive and -negative entities, as was achieved in other studies, was not possible in this study. However, the ORR of 20% is similar to that seen in the BA studies for HPV-negative disease, which suggests a similar pattern of efficacy for this bifunctional agent as BA, assuming the analysis is carried out within a majority HPV-negative cohort.

Finally, a phase I/II clinical trial (NCT04247282) studied the use of neoadjuvant BA on patients who had either node-positive or stage IV disease [78]. There were fourteen total participants in this study. Eight participants were male and six were female, a majority of which identified as white. In addition, 10/14 participants had stage IV disease. Participant tumors were biopsied on day one of the study, which was followed by two doses of BA over two weeks, then followed by surgical resection. Researchers sent the biopsies and post-treatment surgical resections for analysis by sequencing, multispectral immunofluorescence, and antigen-specific T-cell responses. Notably, 12/14 received two doses of BA before surgery, while 2/14 only received one dose. Pathological assessment after surgical resection showed a PR in 5/14 (36%) tumors, as assessed through the Immune Related Pathologic Response Criteria (irPRC), which include the presence of infiltrating macrophages and lymphocytes, tumor cell death–cholesterol clefts, and tissue repair neovascularization and fibrosis [99,100]. Analysis by multispectral immunofluorescence showed a reduction in Ki67, a key marker of cell proliferation, within the primary tumor of those that had a partial response. In addition, the surrounding stroma of these tumors exhibited increased proliferation, thought to signify an increase in immune activation. RNA sequencing analysis of samples before and after treatment interestingly showed inconsistent responses of TGF-β-related signaling factors, which will be discussed later in this review. These results suggest that there is an axis of immune activation and tumor proliferation guiding response to BA, which may not be intrinsically linked to the direct impact on TGF-β signaling levels by BA. It is difficult to compare the results of this study to the others seen within BA, given the use of irPRC in determining treatment response. Again, this study only looks at HPV-negative cancers, thereby limiting its ability to identify unique programs guiding improved treatment response to BA within this tumor entity. 

### 5.3. Joint TGF-β and EGFR Antagonism

In addition to immune checkpoint inhibitors, anti-EGFR antibodies such as cetuximab are used in the setting of metastatic and recurrent illness. However, the efficacy of these treatments is low due to multiple mechanisms of therapy resistance [101], many of which are linked to the dysregulation of the TGF-β pathway. Specifically, it has been shown that elevated levels of TGF-β in the TME are associated with immune evasion and resistance to cetuximab therapy [102]. In fact, in HNSCC patients, a TGF-β-mediated increase in regulatory T cells has been shown to correlate with resistance to cetuximab and a worse overall prognosis [103,104]. The secretion of TGF-β by CAFs has also been shown to be a key mechanism of therapy resistance due to the induction of EMT [105]. TGF-β can also induce EMT through its transactivation of EGFR, synergistically leading to increased tumor invasion and therapy resistance [106]. 

Recent efforts have been made to combine TGF-β and EGFR antagonists to address some of these mechanisms of therapy resistance. BCA101, a novel bifunctional agent consisting of an anti-EGFR IgG1 monoclonal antibody fused to the extracellular domain of TGF-β, has been shown through preclinical data to target TGF-β and EGFR within the TME [107]. BCA-101 effectively downregulated regulatory T cells, reduced EMT, and reduced overall tumor size in a mouse xenograft model of HNSCC [107]. These results were amplified with the addition of a PD-L1 inhibitor in the in vitro setting [107]. Building on these results, BCA-101 was brought into clinical trials in a phase one dose-escalation trial to treat advanced solid tumors refractory to standard therapies, either as monotherapy or in combination with pembrolizumab (NCT04429542). Early preliminary clinical results presented at the 2022 American Society of Clinical Oncology (ASCO) annual meeting detailed the responses of 60 patients, with 6 HNSCC patients receiving monotherapy and 7 HNSCC patients receiving combination therapy [80]. For single-agent therapy, 1/40 (2.5%) participants had a partial response and 13/40 (33%) achieved stable disease. However, for combination therapy, 4/13 (31%) had a partial response and 9/13 (69%) had disease control. Within this combination cohort, 2/7 (29%) HNSCC participants had a partial response, which includes one individual who was refractory to anti-PD-L1 and cetuximab therapy. Currently, this trial is undergoing clinical expansion into HNSCC, with comprehensive results forthcoming. While it may be too soon to draw distinct conclusions on these data, it appears that there is a moderate improvement in ORR within the combination cohort relative to those results seen within BA. However, this study is limited in its numbers of participants, and thus, more study participants are needed to further explore the efficacy of this therapy. 

### 5.4. ALK-1 Receptor Inhibition

There are several agents in both preclinical and clinical development that have been shown to disrupt TGF-β signaling through alternative pathways influencing tumor angiogenesis. The most rigorously studied agent of these is dalantercept, an Fc-fusion protein to ALK-1, shown to have high affinity for BMP-9 and BMP-10 but surprisingly low affinity for TGF-β [72]. As such, through the antagonism of pro-angiogenic BMP-9- and BMP-10-related signaling, this therapy may push the TME to favor blood vessel maturation (Figure 3) [68]. Early preclinical data support this mechanistic proposal as dalantercept demonstrated an efficient ability to antagonize the ALK-1 pathway, as well as reduce tumor burden and vascularity in a mouse model of breast cancer [72]. Further studies showed the benefit of dalantercept in reducing microvascularity and improving the effect of chemotherapy in HNSCC [73,108]. Taken together, these results suggest strong preclinical potential for dalantercept as an anti-tumorigenic agent in HNSCC.

Given early success in preclinical studies, dalantercept was moved into clinical trials. A phase I dose-escalation trial in patients with advanced solid tumors (NCT00996957) included three patients with HNSCC in the final analysis, with 1/3 (33%) achieving a partial response and 1/3 (33%) achieving prolonged stable disease [81]. In a phase II multicenter multiple-dose clinical expansion of this study (NCT01458392), 40 patients with recurrent or metastatic HNSCC naïve to antiangiogenic therapy were eligible for analysis [82]. Most of these participants were men around the age of 60. Roughly half the participants were HPV-positive. Two participants received a fixed dose of 80 mg every three weeks, and then, the study protocol was amended to randomize participants into 0.6 mg/kg or 1.2 mg/kg weight-based dosing every three weeks. The 80 mg fixed participants were excluded from the final efficacy analysis. Of those in the final analysis, 2/40 (5%) achieved a partial response and 14/40 (35%) achieved stable disease, with five participants having disease control for at least three months. All partial responses occurred within the 1.2 mg/kg dosing regimen. Like the results from the BA trials, those participants with HPV-positive tumors had a higher disease control rate than those who were HPV-negative (55% vs. 20% of participants). Notably, this study was limited in its analysis due to problems titrating the dosing regimen. Not only were two participants’ data excluded, due to the initial fixed dosing regimen, but the 0.6 mg/kg dosing regimen failed to produce a partial response in any of the study participants and introduced more heterogeneity. This limited the statistical power to draw any conclusions on significant biomarkers. As such, while dalantercept may have overall poor clinical efficacy in this population, the ORR of 5% is likely an underestimation. The increased disease control rate within HPV-positive cohorts affirms the preferential response that HPV-positive HNSCC has to TGF-β targeting agents, as was also seen in the BA and BCA101 clinical trials. 

## 6. Biomarkers and Patient Stratification

While early preclinical experiments showed promising results for TGF-β targeting agents, the data from clinical trials reflect an ongoing struggle to actualize these benefits into clinical practice. Many of these studies have come up against fundamental limitations regarding participant enrollment, with the limited representation of patients with HNSCC among broader advanced solid tumor participants. Further, many of these studies lack diversity in participant representation. Most participants within these studies are men, which may reflect the epidemiological distribution of this disease, as men are at a two- to four-fold increased risk of developing HNSCC; however, this may limit the generalizability of study findings to the broader population [1]. Additionally, while many studies do not report race and ethnicity, those that do have been composed of mostly Caucasian men, which may limit the capacity to draw inferences from genetic polymorphisms or disease expression programs that may present differently in disparate ancestral lineages. 

Perhaps most importantly, many of these studies have fallen short in their ability to effectively identify participants who may preferentially respond to TGF-β targeting therapy. These struggles with patient stratification also present in the context of significant underlying mechanisms of therapy resistance even within those that do respond to therapy. Given the context-dependent role of TGF-β within the tumor microenvironment, as either a tumor suppressor or a tumor promoter, it is likely that multiple mechanisms of therapy resistance are limiting the clinical utility of this targeted therapeutic strategy. These challenges warrant the use of biomarkers to predict therapy response and combinatorial treatments to overcome mechanisms of therapy resistance. As such, the remainder of this review will focus on the advancements in biomarkers used for stratifying patients for treatment with TGF-β targeting therapy. Of note, a large majority of the current literature focuses on the use of biomarkers within patients treated with BA, and thus, discussion will center around those findings. 

### 6.1. TGF-β Signaling

Perhaps most relevant to the study of these agents is the stratification of responders by relative TGF-β-related signaling. This includes the expression of TGF-β before and after treatment, as well as the related levels of downstream factors such as SMAD-related proteins. However, the connection between clinical response and relative levels of TGF-β signaling is not clear cut. Researchers from the phase I BA study analyzed HNSCC primary tumor samples post-treatment to evaluate expression of TGF-β-related signaling profiles. Even so, little was mentioned regarding the impact of TGF-β-related transcripts and response to BA [76]. Further, within the neoadjuvant trial treating HNSCC patients with BA, whole-exome sequencing data showed no relationship between TGF-β signaling gene mutations and the likelihood of a partial response [78]. Multispectral immunofluorescence (MIF) analysis also showed inconsistent patterns of TGF-β-related gene expression changes when comparing pre-and post-treatment tumor biopsies. Their analysis found that only a small cohort exhibited the expected downregulation of phosphorylated SMAD2 and phosphorylated SMAD3 after neoadjuvant treatment with BA [78]. These results suggest a decoupling of the treatment response and relative expression of TGF-β-related proteins. In addition, within the phase I trial for SHR-1701, aggregated p-SMAD2 levels across all cancer subtypes (advanced solid tumors) showed a trend towards increased ORR with high baseline SMAD-2 levels [79]. It is unclear whether this trend would hold true within an HNSCC subgroup analysis. In fact, within patients with HPV+ cancers (with a majority of cervical, head and neck, and anal SCC), lower baseline TGF-β1 levels were associated with a clinical response to BA [109]. As such, while there are mixed results in the utility of TGF-β signaling in predicting clinical response in HPV-negative disease, it is possible that baseline TGF-β1 levels may serve as a predictor of clinical response in HPV-positive disease. 

### 6.2. Immune Characterization

The relationship between PD-L1 expression and response to immunotherapy within HNSCC is well studied, as a combined positive score >1 has been shown to be associated with therapy response [13,16,17]. Importantly, this biomarker does not catch all responders, as many individuals with a CPS < 1 also show clinical response [110,111]. Interestingly, the utility of tumor PD-L1 expression is less apparent within patients treated with BA. In the phase I trial of BA, there was no correlation between tumor PD-L1 expression and treatment response [76]. Within the neoadjuvant trial of BA, similarly, there was no correlation between pretreatment tumor PD-L1 expression and clinical response. As such, PD-L1 may not be reliable biomarker of therapy response for BA, thereby necessitating further immune characterization of treatment responders. 

Some early work has been carried out to further characterize immune correlates to therapy response within HNSCC patients treated with BA. Single-cell RNA-sequencing data from murine models of HNSCC treated with BA showed preferential response within tumors containing high infiltration of CD8+ and NK cells, along with high expression of MHC class I, CXCL9/10, and CCL5 [95]. Building on these data, the neoadjuvant trial of BA in HNSCC found that high expression of tumor cell MHC class I was independently predictive of a partial response to therapy, even though PD-L1 expression was not [112]. This thereby positions MHC class I as a potential biomarker to predict therapy response. Further, after culturing tumor infiltrating lymphocytes (TILs) before and after neoadjuvant BA, this trial showed that BA was able to induce neoepitope specific T-cell responses in 19/24 (80%) patients [78]. They additionally found that both the number and cumulative magnitude of the neo-epitope-specific response was associated with a PR in the primary tumor. These results suggest that both the presence and functionality of T cells before treatment can serve as an important biomarker in predicting therapy response. As such, to better identify patients who may response to treatment with BA, future studies should assess the number of TILs before enrollment. 

### 6.3. HPV Status

It is important to acknowledge the large impact that HPV status has played in response to TGF-β targeting therapies. It is well known that HPV positivity is associated with a better prognosis in HNSCC [113]. Preclinical data have shown that HPV+ HNSCC patients respond better to radiation [114]. This is thought to be secondary to impaired DNA damage repair and improved immune activation within HPV+ disease [115,116]. HPV status has also been closely linked to response to immunotherapy, with HPV+ patients exhibiting a more robust response to PD-L1 inhibitors [17,117]. In fact, a systemic review and meta-analysis showed that patients with HPV+ HNSCC are 1.29 times more likely to respond to immunotherapy and have a 2-fold increase in overall survival [118]. Previous efforts to further characterize the nature of HPV+ oropharyngeal and cervical cancer also identified dysregulated TGFBRI/TGF-β signaling as highly implicated in the pathogenesis of this disease [65]. Studies in HNSCC have shown that the downregulation of TGF-β signaling in HPV-positive disease leads to the reduction in a key miRNA (miR-182) [119]. This reduction leads to tumor reliance on DNA repair by error-prone alternative end joining, thereby increasing sensitivity to cytotoxic therapy. As mentioned, previous results implicate low baseline TGF-β1 levels in HPV-positive HNSCC as predictive of response to BA [109]. Interestingly, this may suggest that decreased TGF- β signaling may confer increased sensitivity to immunotherapy within HPV-positive HNSCC. 

Within HPV-positive HNSCC, it is important to note that there are gene signatures associated with response to therapy, specifically for BA. Within the phase II clinical expansion of BA to HPV+ malignancies, including HNSCC, a peripheral immunome before and after treatment showed distinct populations of responders and non-responders [109]. Responders showed an increase in T-cell activation with a higher sCD27/sCD40 ratio and a higher CD8+T cell/myeloid-derived suppressor cell ratio, both suggesting immune activation. In addition, this study identified a panel of factors of which low baseline levels were associated with a clinical response to BA. These factors included CD40L, FGF, HGF, TGF-β1, and many others, with enrichment in pathways regulating EMT. Additionally, after the first round of treatment with BA, responders were less likely to show an upregulation of the immunosuppressive cytokine IL-8. Interestingly, this cytokine has previously been described as an important contributor to tumor proliferation, EMT, angiogenesis, and therapy resistance in multiple types of cancer [120]. As such, it is possible that alternative factors such as IL-8 are driving therapy resistance and decreasing the utility of TGF-β inhibitors for HNSCC. This positions IL-8 as an important biomarker for therapy response as well as a target for combination therapy with TGF-β inhibitors. Taken together, the results of these studies suggest that HPV-positive HNSCC is accompanied by a unique immune-oncogenic profile of factors which can be associated with response to BA. Further work is needed to explore the utility of this profile in a broader patient population. In addition, future studies combining TGF-β inhibition with IL-8 inhibition may help to understand methods of therapy resistance and improve patient outcomes. 

## 7. Conclusions

TGF-β is an important cytokine in the maintenance of tissue homeostasis and cellular migration. Its pleiotropic characteristics in tissue dynamics and tumorigenesis have made it an attractive candidate for clinical trials. However, despite encouraging pre-clinical results, TGF-β inhibitors have yet to be approved for clinical practice in the treatment of HNSCC. Combinatorial treatment, synthesizing TGF-β antagonists along with other traditional therapies, such as PD-L1 or EGFR inhibition, has shown variable success, with an emphasis on the promising results seen in HPV+ individuals. While preliminary biomarker analysis has identified the importance of the immune microenvironment in response to therapy, more work is needed to integrate these findings into clinical practice to effectively identify individuals who will respond to therapy. Ultimately, TGF-β inhibition remains a potentially viable approach for the management of recurrent and metastatic HNSCC; however, gaps in knowledge regarding patient stratification will continue to limit its applicability to clinical practice. 

## Figures and Tables

**Figure 1 cancers-16-03047-f001:**
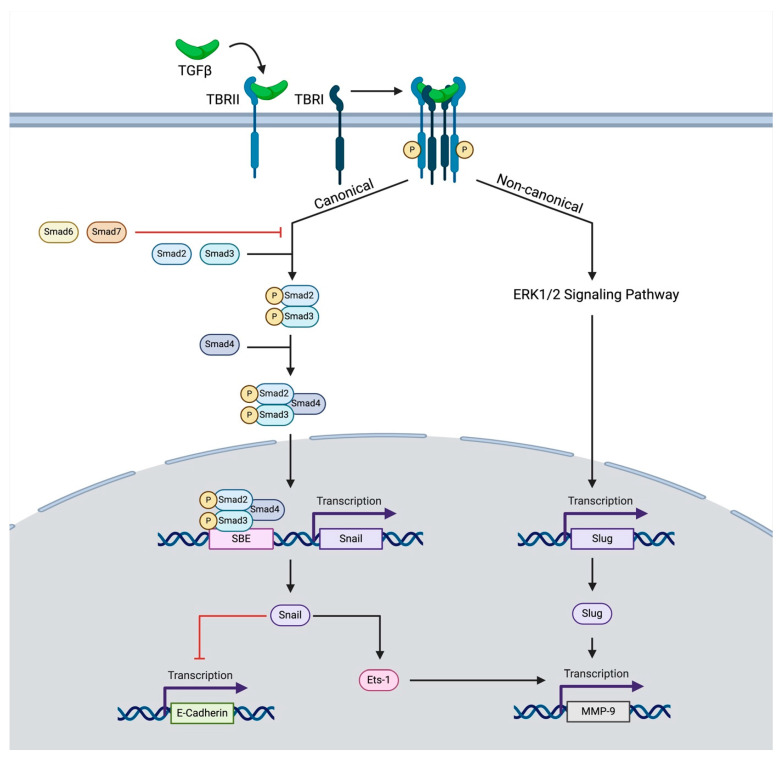
The schematic depicts canonical and non-canonical TGF-β signaling influencing EMT. Upon binding to its receptor, TBRII forms a heterotetrameric complex leading to the phosphorylation of TBRI. This induces canonical signaling, in which SMAD-related proteins lead to the transcription of SNAIL-1, which either inhibits the transcription of E-Cadherin or promotes the transcription of MMP-9. The transcription of MMP-9 is also promoted through a non-canonical pathway in which ERK1/2 signaling-induced SNAIL2 (SLUG) transcription further augments MMP-9 transcription.

**Figure 2 cancers-16-03047-f002:**
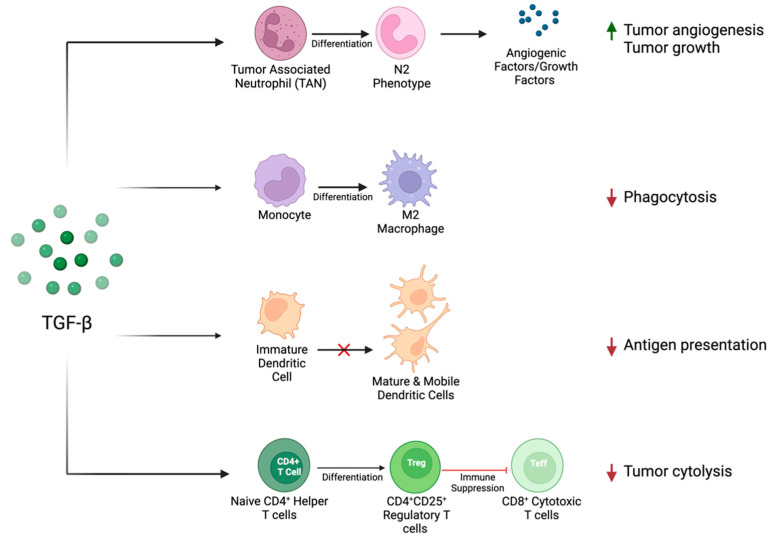
TGF-β influences the immune–tumor interface to create a pro-oncogenic TME through its action on tumor-associated neutrophils, monocytes, dendritic cells, and lymphocytes. This action promotes tumor growth and angiogenesis and suppresses the immune-related detection and destruction of tumor cells.

**Figure 3 cancers-16-03047-f003:**
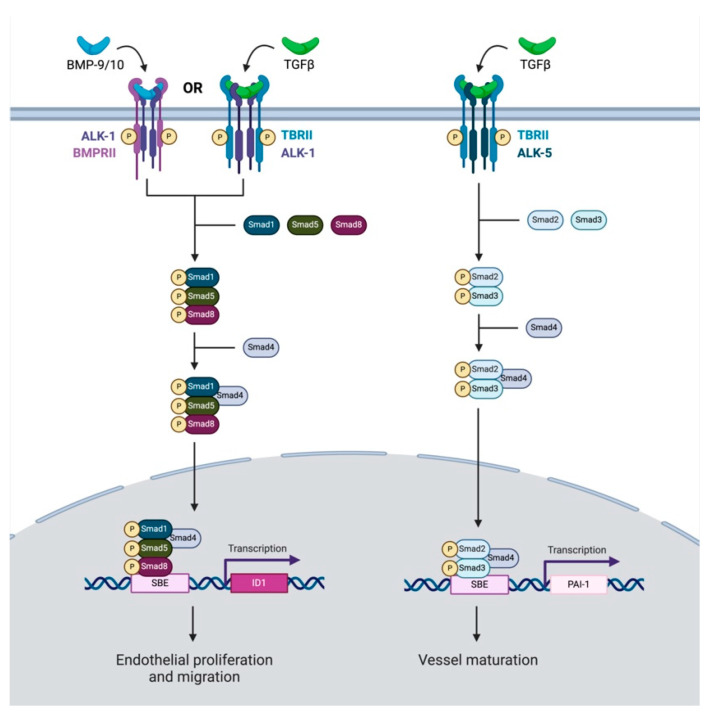
This schematic reflects TGF-β signaling pathways which influence angiogenesis. TGF-β may either have a pro-angiogenic or anti-angiogenic influence on cancer. TGF-β or BMP-9/10 attach to ALK-1 or BMPRII, respectively. Downstream SMAD-1, -5, and-8 related signaling coupled with SMAD-4 leads to the transcription of ID1, which promotes endothelial cell proliferation and migration. Conversely, TGF-β can also bind to an ALK-5 receptor. This initiates downstream signaling with Smad-2 and Smad-3, which similarly couple with Smad-4. However, this complex translocates to the nucleus and leads to the transcription of PAI-1, promoting vessel maturation.

**Figure 4 cancers-16-03047-f004:**
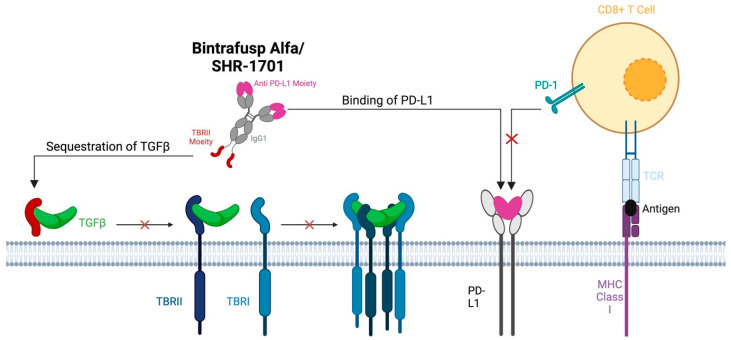
Both bintrafusp alfa and SHR-1701 consist of an IgG1 anti-PD-L1 component fused to the extracel-lular domain of TGF-BRII. These therapies function by competitively inhibiting the PD-L1 receptor on tumor cells and by binding serum circulating TGF-β to reduce TGF-β-related signaling.

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
