# Peer review of "Advancements in TGF-β Targeting Therapies for Head and Neck Squamous Cell Carcinoma"

_cancers, 2024, doi:10.3390/cancers16173047_

Round 1
Reviewer 1 Report
Comments and Suggestions for Authors
This review provides a solid foundation for understanding the role of TGF-β in HNSCC and highlights the need for more targeted therapeutic approaches, including patient stratification. Addressing the weaknesses and incorporating the suggested improvements will enhance the review's impact and utility for researchers and clinicians in the field.
1) Provide a more detailed analysis of clinical trials involving TGF-β inhibitors, including a comparison of different inhibitors, trial designs, patient demographics, and specific reasons for suboptimal outcomes.
2) Include more detailed discussions on the molecular mechanisms of TGF-β signaling, particularly how it interacts with other pathways like EGFR, and its role in the tumor microenvironment.
3) Propose potential biomarkers or genomic profiles that could be used to stratify patients for TGF-β inhibitor therapy. Discuss ongoing research or existing data supporting these biomarkers.
4) Outline future research directions, including potential combination therapies with TGF-β inhibitors and other targeted agents or immunotherapies, and how these combinations might overcome current limitations.
Author Response
Comment 1: Provide a more detailed analysis of clinical trials involving TGF-β inhibitors, including a comparison of different inhibitors, trial designs, patient demographics, and specific reasons for suboptimal outcomes.
Response 1:
Thank you for this comment. We agree that more information is needed to add context to the clinical studies presented in this study. As such, we have gone through the clinical studies and added more information about the study designs, patient demographics, and limitations. We have additionally tried to add brief comparisons to the other TGF-B targeting agents and immunotherapy throughout this section. All of these changes can be found in our discussion of the clinical studies on TGF-β inhibitors.
Comment 2: Include more detailed discussions on the molecular mechanisms of TGF-β signaling, particularly how it interacts with other pathways like EGFR, and its role in the tumor microenvironment.
Response 2:
Thank you for pointing out the need to further explore the interactions between the EGF pathway and TGF-β signaling. We agree that this would help our paper and add further context to the use of BCA101 as a clinical therapy. As such, we have added more information in section 4.1 (lines 193-200) on the connection between TGF-β and EGF through their shared influence on Notch signaling mediated cell differentiation and EMT.
Comment 3: Propose potential biomarkers or genomic profiles that could be used to stratify patients for TGF-β inhibitor therapy. Discuss ongoing research or existing data supporting these biomarkers.
Response 3:
Thank you for this suggestion. We agree that this section of our paper needs more directive biomarker proposals to improve future studies. As such, we have adapted the three biomarker subsections to incorporate this feedback. In the TGF-β signaling section we adjusted the end to propose TGF-β1 as a biomarker exclusively within HPV-positive disease (line 596-599). Within the immune characterization section, we added more information on the role of tumor infiltrating lymphocytes and neoepitope specific T-cell responses in predicting response to BA. We have proposed measuring tumor infiltrating lymphocytes before study enrollment to better identify potential treatment responders (Lines 618-624). We have also proposed baseline MHC class I as a potential biomarker for treatment response (lines 617-618). Within the HPV status section, we included more information about the connection between TGF-β signaling and HPV status (lines 639-645). We also provided a more thorough analysis of the Tsai et al. 2022 paper and presented many factors associated with clinical response to BA (lines 653-654). Finally, we proposed IL8 both as a potential biomarker and combinatorial therapeutic target to improve future studies for TGF-β inhibitors (lines 656-661).
Comment 4: Outline future research directions, including potential combination therapies with TGF-β inhibitors and other targeted agents or immunotherapies, and how these combinations might overcome current limitations.
Response 4:
We appreciate your suggestion to further explore the limitations of previous clinical studies and suggest future directions. To address this, we added more information regarding the current limitations associated with TGF-β inhibitor clinical studies before discussing specific biomarkers (lines 551-573). Per your previous comment, we also developed our section on biomarkers and patient stratification. Given the relevance of IL-8 expression in predicting response to therapy in HPV-positive HNSCC, we proposed combining anti-IL8 therapy with TGF-β inhibition (Lines 658-661). After acknowledging the importance of T-cell neo-epitope response in HNSCC, we also proposed assessing infiltrating tumor lymphocytes before study participation to identify patients who may respond best to TGF-β targeting therapy (lines 623-626).
Reviewer 2 Report
Comments and Suggestions for Authors
The manuscript provides a thorough background on TGF-β signaling and its implications for HNSCC. It includes both preclinical and clinical data, offering a balanced perspective on the current status of TGF-β inhibitors. The document is well-organized, with clear sections that guide the reader through the complex topic. Overall, the manuscript is well-researched and provides valuable insights into the role of TGF-β in HNSCC. Addressing the above suggestions will enhance its clarity, impact, and usefulness for readers and researchers in the field.
Major comment:
1. The introduction provides a good overview but can be streamlined to focus more on the unmet clinical need for targeted therapies in HNSCC. Including more statistics on the current success rates of existing therapies would strengthen the argument.
2. The review of clinical trials could be more detailed. Summarizing the key outcomes, patient populations, and specific challenges faced in each trial would provide a clearer picture of the clinical landscape. It would be helpful to include a table summarizing the main findings of preclinical and clinical studies on TGF-β inhibitors.
3. The discussion section should include a more critical analysis of why certain clinical trials may have failed and what could be done differently in future studies.
4. Potential biomarkers for patient stratification should be discussed in more detail. This would provide actionable insights for future research.
Specific Suggestions:
1. Page 1, Line 1-2: “Current Progress in the use of TGF-β Targeting Agents for Head and Neck Squamous Cell Carcinoma” - Consider rephrasing to: “Advancements in TGF-β Targeting Therapies for Head and Neck Squamous Cell Carcinoma.”
2. Page 1, Line 12: “Simple Summary: TGF-β is an important cytokine implicated in the pathogenesis of head and neck squamous cell carcinoma (HNSCC).” - Could be expanded to briefly explain why TGF-β is significant and its dual role.
3. Page 2, Line 36-37: “This cancer includes malignant lesions which arise within the oral cavity, nasal cavity, and salivary glands…” - Consider adding a sentence about the prognosis and survival rates associated with HNSCC to underscore the clinical importance.
4. Page 2, Line 49-50: “Chemotherapy typically consists of a platinum-based regimen, which is often accompanied by significant toxicity…” - Could add a specific example of such toxicities to illustrate the severity.
5. Page 3, Line 75-76: “This program, termed partial epithelial-to-mesenchymal transition (p-EMT), confers metastable epithelial characteristics onto highly mobile mesenchymal cells…” - Consider simplifying the language for clarity.
6. Page 6, Line 126-128: “In epithelial tissue and early stages of oncogenesis TGF-β acts as a tumor suppressor…” - This section could be expanded to explain the mechanisms in simpler terms for clarity.
Author Response
Comment 1: The introduction provides a good overview but can be streamlined to focus more on the unmet clinical need for targeted therapies in HNSCC. Including more statistics on the current success rates of existing therapies would strengthen the argument.
Response 1:
Thank you for pointing out the need to strengthen this part of our paper. We agree that adding more information on HNSCC survival and treatment efficacy could strength our argument about the unmet clinical need in metastatic and recurrent HNSCC. As such, we have added more information about the survival rates of HNSCC as outlined in the 2009 SEER report (Lines 47-50). We have also added information about survival from early-stage disease with minimal nodal involvement and compared this to survival from stage III or stage IV unresectable disease (lines 58-63). Finally, we added information regarding the survival of patients with recurrent or metastatic disease treated with cetuximab monotherapy to show how survival outcomes worsen after immunotherapy options are exhausted (lines 69-71).
Comment 2: The review of clinical trials could be more detailed. Summarizing the key outcomes, patient populations, and specific challenges faced in each trial would provide a clearer picture of the clinical landscape. It would be helpful to include a table summarizing the main findings of preclinical and clinical studies on TGF-β inhibitors.
Response 2:
Thank you for this comment. We agree that our study could benefit from a more critical overview of the clinical trials discussed. As such, we have gone through and added additional information regarding study design, patient demographics, dosing regimens and study limitations. These changes can be found throughout our discussion of the clinical studies on TGF-β inhibitors. Thank you for your suggestion regarding the addition of a table to summarize the results of these studies. Our Table 1 summarizes all the main findings of the clinical papers discussed in this review. The preclinical data on TGF-β inhibitors are limited by the fact that many of the initial preclinical studies were not done specifically on models of HNSCC. Additionally, given the heterogeneity of how therapy response is measured across preclinical studies, we believe a textual discussion of these studies, rather than a table, may be the best way to discuss the relevant results. As such, through our discussion, our study attempts to capture some of the key preclinical findings associated with each TGF-β inhibitor that prompted advancement to clinical trials.
Comment 3: The discussion section should include a more critical analysis of why certain clinical trials may have failed and what could be done differently in future studies.
Response 3:
We appreciate your suggestion to further explore the limitations of previous clinical studies and suggest future directions. To address this, we added more information regarding the current limitations associated with TGF-β inhibitor clinical studies before discussing specific biomarkers (lines 551-573). Some of these limitations include participant recruitment, study size, patient stratification and therapy resistance. We’ve attempted to integrate these limitations into our discussion of each clinical study. We also developed our section on patient stratification to provide suggestions for future directions in clinical studies. Specifically, given the importance of T-cell neo-epitope response in HNSCC, we proposed assessing infiltrating tumor lymphocytes before study participation to identify patients who may respond best to TGF-β targeting therapy (lines 623-626). To address potential mechanisms of therapy resistance we also suggested combining TGF-β inhibiting therapy with anti-IL-8 therapy within HPV-positive HNSCC (Lines 658-661).
Comment 4: Potential biomarkers for patient stratification should be discussed in more detail. This would provide actionable insights for future research.
Response 4:
Thank you for this suggestion. We agree that this section of our paper needs more directive biomarker proposals to improve future studies. As such, we have adapted the three biomarker subsections to incorporate this feedback. In the TGF-β signaling section we adjusted the end to propose TGF-β1 as a biomarker exclusively within HPV-positive disease (line 596-599). Within the immune characterization section, we added more information on the role of tumor infiltrating lymphocytes and neoepitope specific T-cell responses in predicting response to BA. We have proposed measuring tumor infiltrating lymphocytes before study enrollment to better identify potential treatment responders (Lines 618-624). We have also proposed baseline MHC class I as a potential biomarker for treatment response (lines 617-618). Within the HPV status section, we included more information about the connection between TGF-β signaling and HPV status (lines 639-645). We also provided a more thorough analysis of the Tsai et al. 2022 paper and presented many factors associated with clinical response to BA (lines 653-654). Finally, we proposed IL8 both as a potential biomarker and combinatorial therapeutic target to improve future studies for TGF-β inhibitors (lines 656-661).
Specific Suggestions:
Comment 1: Page 1, Line 1-2: “Current Progress in the use of TGF-β Targeting Agents for Head and Neck Squamous Cell Carcinoma” - Consider rephrasing to: “Advancements in TGF-β Targeting Therapies for Head and Neck Squamous Cell Carcinoma.”
Response 1: Thank you for this suggestion. We agree that this revised title more accurately captures the content of this paper. We have changed it accordingly.
Comment 2: Page 1, Line 12: “Simple Summary: TGF-β is an important cytokine implicated in the pathogenesis of head and neck squamous cell carcinoma (HNSCC).” - Could be expanded to briefly explain why TGF-β is significant and its dual role.
Response 2: Agreed. I have adjusted the simple summary to briefly touch upon the importance of TGFB and its dual role in normal tissue and cancer (lines 15-16).
Comment 3: Page 2, Line 36-37: “This cancer includes malignant lesions which arise within the oral cavity, nasal cavity, and salivary glands…” - Consider adding a sentence about the prognosis and survival rates associated with HNSCC to underscore the clinical importance.
Response 3: Thank you for this point. I have included data from the Surveillance, Epidemiology, and End Results (SEER) Program of the U.S. National Cancer Institute analysis from 2009 to further clarify trends in five-year survival for HNSCC. I also added supporting information regarding the reduction in survival for metastatic disease across oral cavity, tongue and tonsillar cancers (lines 47-54).
Comment 4: Page 2, Line 49-50: “Chemotherapy typically consists of a platinum-based regimen, which is often accompanied by significant toxicity…” - Could add a specific example of such toxicities to illustrate the severity.
Response 4:
I agree that it is important to add examples of these toxicities. As such, we added some of the most important severe toxicities associated with platinum-based regimens to the end of this sentence (lines 57-58).
Comment 5: Page 3, Line 75-76: “This program, termed partial epithelial-to-mesenchymal transition (p-EMT), confers metastable epithelial characteristics onto highly mobile mesenchymal cells…” - Consider simplifying the language for clarity.
Response 5: Thank you for this suggestion. I agree that the words “metastable” and “highly mobile” add unnecessary complication to this sentence. I have eliminated them for the sake of clarity (lines 92-95).
Comment 6: Page 6, Line 126-128: “In epithelial tissue and early stages of oncogenesis TGF-β acts as a tumor suppressor…” - This section could be expanded to explain the mechanisms in simpler terms for clarity.
Response 6: Thank you for pointing out the need to clarify this section of our paper. We agree that this section would benefit from additional description and simplification. We have added relevant information describing the proteins involved in the tumor suppressive role of TGF-B. We have also added more clarification on the techniques used induce TGF-β1 expression with Ru486 to describe the tumor promoting role of this cytokine in HNSCC. We have also tried to clarify what was meant by “reactive epithelial changes” induced by TGFB signaling in its tumor promoting role and replaced the phrase “cell autonomous manner” with “through its impact on other cells in the tumor microenvironment” (lines 155-164).
Reviewer 3 Report
Comments and Suggestions for Authors
This review provided a comprehensive review of current knowledge regarding TGF-β targeting therapies in HNSCC. It covered the biological basis of TGF-β signaling, evaluated the effectiveness of various therapeutic agents, and addressed the challenges in clinical implementation. It also discusses the role of biomarkers in predicting response and the impact of HPV status on therapy outcomes as well as the importance of patient stratification for employing TGF-β. Such information and discussion could benefit future research and clinical practice by identifying key factors that influence the efficacy of TGF-β targeting therapies and suggesting areas for further investigation.
1. Lines 20-23: This sentence appears lengthy and convoluted. Please consider revising it to improve readability.
2. Lines 35-37: I believe this statistic is from 2018, not representative of each year. Please consider updating this statistic to more recent data.
3. Line 41: The word “anemia” should not be capitalized.
4. Lines 53-54&58-59&87: The authors repeatedly mention 'metastatic and recurrent disease,' but this lacks specificity. Please clarify whether this refers specifically to metastatic and recurrent HNSCC or various types of cancers.
5. Line 60: The phrase “this disease” appears inappropriate as HNSCC is not a single disease.
6. Lines 60-63: This sentence appears confusing as the phrase “as well as disruption in p53 and RB1” seems tacked on without clearly connecting to the earlier part of the sentence. Please consider revising it for better clarification.
7. Line 74: Both the full term and its abbreviation of “cancer associated fibroblasts (CAFs)” have appeared twice so far. In academic writing, it is generally recommended to spell out the full term followed by the abbreviation in parentheses when introducing it for the first time. Once an abbreviation is introduced, use it consistently throughout the manuscript. Please consider conducting thorough proofreading to remove this issue.
8. Lines 71-75&78-82: This sentence appears lengthy and convoluted. Please consider breaking it down.
9. Line 93: The word “signaling” is misspelled.
10. Line 110: The word “heterotetrameric” is misspelled.
11. Line 126: The phrase "TGF-β serves a bifunctional role in normal tissue and cancer" could be more specific. It would be helpful to briefly outline what these bifunctional roles are.
12. Line 133: Please add the full term for “ATM”.
13. Line 137: Please remove the space between “TGF-β” and “in”.
14. Line 138: “Ru486 induced TGF-β1 signaling” appears unclear. Ru486 is a glucocorticoid receptor (GR) antagonist. Please make sure to briefly introduce what Ru486 is to avoid confusion and ensure readers understand its relevance.
15. Line 159: The citation number was placed outside of the sentence.
16. Lines 167-171: Please consider breaking down this sentence.
17. Line 173: The word “heterotetrameric” is misspelled.
18. Line 182: Please remove the word “a”.
19. Line 184: Please reformat the word “TGF-β Induced” to “TGF-β-induced”.
20. Line 198: Please move the citation mark “[55]” before the comma.
21. Line 214: Please remove the citation mark “[59]”.
22. Line 218: Please remove the word “however”.
23. Line 234: The word “signaling” is misspelled.
24. Line 239: Missing a period at the end of the sentence.
25. Line 231: The term "bFGF" should be spelled out at least once with the abbreviation in parentheses for clarity.
26. Line 254: Could the authors please put citation marks of the specific clinical trials discussed inside the table? This would allow readers to better locate the sources of the data and results.
27. Line 256: The term "Small Molecule Inhibitors (SMIs)" is used without defining it clearly at first. Ensure that SMIs are well-defined for clarity.
28. Line 271: Please move the citation mark “[69]” before the comma.
29. Line 292: The word “illness” appears inappropriate in the context of chemotherapy. Please consider replacing it.
30. Lines 309-312: Please consider placing the citation marks more appropriately.
31. Lines 318-320: Please consider reconstructing this description to improve readability.
32. Lines 332-333: This sentence should be improved for readability such as “four attained a PR, and two achieved a PR after initial progression, raising the ORR to 6/15 (40%).”
33. Line 359: Please place the citation mark properly.
Comments on the Quality of English Language
This manuscript requires extensive proofreading to correct long and convoluted sentences, add necessary punctuation marks where they are currently missing, and address misspellings and misplaced citation marks.
Author Response
Comment 1: Lines 20-23: This sentence appears lengthy and convoluted. Please consider revising it to improve readability.
Response 1: Thank you for your feedback on this introductory sentence. We agree that restructuring this sentence would improve readability. We have made our wording more concise and restructured the content into two separate sentences to address your comment (lines 22-25).
Comment 2: Lines 35-37: I believe this statistic is from 2018, not representative of each year. Please consider updating this statistic to more recent data.
Response 2: Thank you for pointing this out. We have consulted the GLOBOCAN database for updated statistics and have found that these data have not changed significantly since 2018. We have updated this section to reference GLOBOCAN data from 2022 (lines 37-39).
Comment 3: Line 41: The word “anemia” should not be capitalized.
Response 3: Thank you for pointing this out. We have corrected this capitalization error (line 44).
Comment 4: Lines 53-54&58-59&87: The authors repeatedly mention 'metastatic and recurrent disease,' but this lacks specificity. Please clarify whether this refers specifically to metastatic and recurrent HNSCC or various types of cancers.
Response 4: Thank you for bringing this to our attention. We have altered the wording to indicate that we are referring specifically to metastatic and recurrent head and neck squamous cell carcinoma. (lines 66-67, 71-72, 73-75)
Comment 5: Line 60: The phrase “this disease” appears inappropriate as HNSCC is not a single disease.
Response 5: Thank you for pointing this out. We agree the phrase “this disease” is unsuitable for describing HNSCC, given that it is not a singular disease. We have thus replaced this phrase with "HPV-positive and HPV-negative HNSCC” to more accurately describe this entity (line 75).
Comment 6: Lines 60-63: This sentence appears confusing as the phrase “as well as disruption in p53 and RB1” seems tacked on without clearly connecting to the earlier part of the sentence. Please consider revising it for better clarification.
Response 6: Thank you for your comment regarding this sentence. We agree that discussing disrupted genes for both HPV positive and negative HNSCC in a singular sentence may be unclear. To address this concern, we have separated these ideas into consecutive sentences to preserve clarity (lines 75-79).
Comment 7: Line 74: Both the full term and its abbreviation of “cancer associated fibroblasts (CAFs)” have appeared twice so far. In academic writing, it is generally recommended to spell out the full term followed by the abbreviation in parentheses when introducing it for the first time. Once an abbreviation is introduced, use it consistently throughout the manuscript. Please consider conducting thorough proofreading to remove this issue.
Response 7: Thanks for bringing this to our attention. After close inspection, the full term and its abbreviation are stated first on line 81. We have thus made sure that all subsequent mentions of cancer-associated fibroblasts are called CAFs. These edits can be identified on lines 90, 94, 179, 185 and 484.
Comment 8: Lines 71-75&78-82: This sentence appears lengthy and convoluted. Please consider breaking it down.
Response 8: Thank you for noticing this. We have broken down this sentence significantly and simplified the language to add clarity to our message (lines 87-90, 94-98)
Comment 9: Line 93: The word “signaling” is misspelled.
Response 9: Thank you for pointing this out. We have corrected the spelling (line 109).
Comment 10: Line 110: The word “heterotetrameric” is misspelled.
Response 10: Thank you, we have adjusted this (line 126).
Comment 11: Line 126: The phrase "TGF-β serves a bifunctional role in normal tissue and cancer" could be more specific. It would be helpful to briefly outline what these bifunctional roles are.
Response 11: Thank you for this suggestion. I have modified the sentence as follows “TGF-β acts as either a tumor suppressor or tumor promoter depending on tissue context” (line 142-143).
Comment 12: Line 133: Please add the full term for “ATM”.
Response 12: Thank you for noting this, we have incorporated the full name of this protein in the manuscript (line 150).
Comment 13: Line 137: Please remove the space between “TGF-β” and “in”.
Response 13: Thank you, we have completed this (line 152).
Comment 14: Line 138: “Ru486 induced TGF-β1 signaling” appears unclear. Ru486 is a glucocorticoid receptor (GR) antagonist. Please make sure to briefly introduce what Ru486 is to avoid confusion and ensure readers understand its relevance.
Response 14: Thank you for this comment. We added an additional citation linked to a study explaining this transgenic technique and added a description of Ru486 in its role as an inducer of a linked TGFB1 transactivator (line 155-159).
Comment 15: Line 159: The citation number was placed outside of the sentence.
Response 15: Thank you for noting. We have fixed this (line 180).
Comment 16: Lines 167-171: Please consider breaking down this sentence.
Response 16: Thank you for this comment. We have separated this phrase into two sentences (line 188-192).
Comment 17: Line 173: The word “heterotetrameric” is misspelled.
Response 17: Thank you for noting. We have adjusted this (line 215).
Comment 18: Line 182: Please remove the word “a”.
Response 18: Thank you for noting. We eliminated this word (line 223).
Comment 19: Line 184: Please reformat the word “TGF-β Induced” to “TGF-β-induced”.
Response 19: Thanks for this suggestion. We have reformatted this for clarity (line 226).
Comment 20: Line 198: Please move the citation mark “[55]” before the comma.
Response 20: Thank you for picking up on this. We have moved the comma to after the citation (line 239).
Comment 21: Line 214: Please remove the citation mark “[59]”.
Response 21: Thank you for catching this. We have removed this citation (line 274).
Comment 22: Line 218: Please remove the word “however”.
Response 22: Thank you for noting. We have removed this word (line 278).
Comment 23: Line 234: The word “signaling” is misspelled.
Response 23: Thank you for catching this. We have changed the spelling of this word (line 294).
Comment 24: Line 239: Missing a period at the end of the sentence.
Response 24: Thank you for noting. We have added a period here (line 299).
Comment 25: Line 231: The term "bFGF" should be spelled out at least once with the abbreviation in parentheses for clarity.
Response 25: Thank you for noting this. I have spelled out this word before abbreviating it (line 291).
Comment 26: Line 254: Could the authors please put citation marks of the specific clinical trials discussed inside the table? This would allow readers to better locate the sources of the data and results.
Response 26: Thank you for this suggestion. We agree that adding citations to this table will help readers to navigate to the associated studies. We have added citations for the major papers for each of these studies which can be found in the second column of our table 1.
Comment 27: Line 256: The term "Small Molecule Inhibitors (SMIs)" is used without defining it clearly at first. Ensure that SMIs are well-defined for clarity.
Response 27: Thank you for pointing this out. We agree that a definition is needed for small molecule inhibitors before we discuss their clinical utility. As such, we have added the sentence, “These inhibitors have a molecular weight of less than 900Da and are used to target specific proteins within cancer cells or the tumor microenvironment” (lines 339-340).
Comment 28: Line 271: Please move the citation mark “[69]” before the comma.
Response 28: Thank you for noting this. We have changed the citation to occur before the comma (line 354).
Comment 29: Line 292: The word “illness” appears inappropriate in the context of chemotherapy. Please consider replacing it.
Response 29: Thank you for pointing this out. I replaced “illness” with "metastatic or recurrent HNSCC” to address this comment (line 382).
Comment 30: Lines 309-312: Please consider placing the citation marks more appropriately.
Response 30: Thank you for catching this. I have adjusted the two misplaced commas accordingly (lines 406-407).
Comment 31: Lines 318-320: Please consider reconstructing this description to improve readability.
Response 31: Thank you for noting this. We agree that this description could be more clear. We have adjusted it as follows: “Both Bintrafusp Alfa and SHR-1701 consist of an IgG1 anti-PD-L1 component fused to the extra-cellular domain of TGF-BRII. These therapies function by competitively inhibiting the PD-L1 receptor on tumor cells and by binding serum circulating TGF-β to reduce TGF-β related signaling” (lines 398-400).
Comment 32: Lines 332-333: This sentence should be improved for readability such as “four attained a PR, and two achieved a PR after initial progression, raising the ORR to 6/15 (40%).”
Response 32: Thank you for noting this. I have adjusted this sentence accordingly to match your suggestion (lines 429-430).
Comment 33: Line 359: Please place the citation mark properly.
Response 33: Thank you for catching this. I have adjusted the location of the comma to correct this (line 479).